# Meat Production Potential of Local Horse Breeds: Sustainable Conservation Through Valorization

**DOI:** 10.3390/ani15131911

**Published:** 2025-06-28

**Authors:** Ante Ivanković, Mateja Pećina, Giovanni Bittante, Nicoló Amalfitano, Miljenko Konjačić, Nikolina Kelava Ugarković

**Affiliations:** 1Department of Animal Science and Technology, University of Zagreb Faculty of Agriculture, Svetošimunska 25, 10000 Zagreb, Croatia; matejapecina@agr.hr (M.P.); mkonjacic@agr.hr (M.K.); nkelava@agr.hr (N.K.U.); 2Department of Agronomy, Food, Natural Resources, Animals and Environment (DAFNAE), University of Padova, Via dell’Università 16, 35020 Legnaro, Italy; bittante@unipd.it (G.B.); nicolo.amalfitano@unipd.it (N.A.)

**Keywords:** horse, local breed, meat quality, fatty and amino acids, sensory valorization

## Abstract

The Croatian Posavina horse is an indigenous cold-blooded breed traditionally reared in the floodplain areas of the Sava River. Although once endangered, the breed has regained population stability through conservation efforts and is now primarily used for pasture-based meat production. This study aimed to evaluate the growth performance, carcass traits, and meat quality of male foals reared under semi-extensive conditions. The results showed that at eleven months of age, the foals achieved a solid live and carcass weight and produced meat with favorable technological and nutritional properties. The meat was high in protein, low in fat, and had a desirable profile of polyunsaturated fatty acids, including a balanced ω-6/ω-3 ratio. Sensory analysis confirmed the meat’s tenderness, pleasant flavor, and attractive appearance. Compared to other European horse breeds, the Croatian Posavina horse demonstrated competitive production traits and superior nutritional quality. These findings highlight the potential of this local breed not only for sustainable meat production but also for contributing to the preservation of agro-biodiversity and rural identity. Promoting its use in niche markets could ensure both economic viability and long-term conservation.

## 1. Introduction

Animal farm genetic resources represent a significant genetic potential of a given area, shaped by environmental, economic, social, and other pressures. In addition to their specific genetic combinations, they are also a component of the identity and cultural heritage of the area, well adapted to maintaining local ecosystems. Besides direct economic benefits, there are numerous indirect benefits of their *in situ* conservation (ecosystem services, regional identity, local gastronomy, etc.). Local horse breeds have come under pressure during the second half of the 20th century due to changes in the economic and social environment, especially the shift in modes of transportation for goods and people, which marginalized certain breeds, particularly cold-blooded ones, to the brink of extinction. Some local breeds have even disappeared. The adaptation of local breeds to the needs of the socio-economic environment is the foundation for their conservation and long-term sustainability. According to the FAO DAD-IS database, the population status of 54.3% of all horse breeds is unknown, while 9.8% are extinct. Of the 720 existing local horse breeds, the majority (52.8%) are in Europe and the Caucasus, but only 7.2% are considered not endangered [1]. Warm-blood and pony breeds are mostly involved in recreational and sports activities, while their commercial use in milk or meat production is less common, partly due to consumer reluctance based on religious, ethical, or traditional beliefs, and partly due to a lack of full information. The consumption of horse meat in Croatia is modest and not widespread, with the exception of Istria County. However, the country’s geographical position, favorable marginal pastures, and existing breeding practices make horse meat production profitable and oriented toward export [2], primarily to the neighboring Italian market, which remains one of the largest consumers of horse meat in Europe. The horse meat scandal involving frozen products across Europe further contributed to public skepticism regarding accurate consumer information. Agnoli et al. [3] highlight that, in the context of the horse meat scandal, European consumers expressed heightened concern over meat authenticity in ready meals, showing a marked preference for nationally sourced products and transparent labeling as key factors in reinforcing trust and preventing food fraud. Self-sustainability is a key element for the safe and long-term preservation of endangered breeds, and it is based on their economic and social valorization, particularly through repositioning their usability. The economic potential of local cold-blooded horse breeds is often assessed through the quantity and quality of their products, as well as other benefits (ecosystem services, intrinsic value, etc.).

A limited number of studies have investigated slaughter traits in various horse breeds [4,5,6,7,8,9,10,11,12,13,14,15,16,17,18,19,20], as well as carcass meat yields [4,5,8,11,14,15,16,17]. Some researchers have extended their investigations to include the physical characteristics of horse meat [13,14,17,18,19,21,22,23,24,25], its chemical composition [9,12,13,14,15,16,17,18,19,22,23,24,25,26,27], fatty acid profile [13,15,18,19,25,28,29,30,31,32], and amino acid profile [18,19,30,33]. Furthermore, studies have addressed the nutritional advantages of horse meat over other types of meat, potential risks during storage [34], and the effects of aging processes on meat quality [30,35]. Given the ethical dilemmas associated with horse meat consumption, some studies have also focused on consumer attitudes [36,37,38,39,40,41,42]. There is clear interest from the scientific community and the broader public in this type of meat, especially when considering the many additional benefits of in situ conservation programs of local horse breeds in their native ecosystems. The Croatian Posavina horse is one of the indigenous horse breeds in Croatia that has, over the past three decades, been brought back from the brink of extinction to a population of approximately 5000 registered breeding adult individuals. It is primarily raised under extensive farming systems in the floodplain areas of the Sava River basin, typically on pastures less suitable for other forms of agricultural production (Appendix A). In protected areas of high ecological value, local horse breeds serve as important ecosystem service agents. Maintaining the Croatian Posavina horse in such valuable habitats through appropriate management models provides economic viability for local farmers and ensures a “win–win” outcome for all integrated stakeholders. To date, research has focused on the phenotype and genetic structure of the breed [43]; however, a comprehensive valuation of the breed’s utility, particularly through a meat production program, is necessary to enable the improved market repositioning of the breed. Therefore, the aim of this study is to evaluate the growth potential, carcass quality, and meat characteristics of the Croatian Posavina horse, providing a basis for optimizing conservation practices.

## 2. Materials and Methods

### 2.1. Animals, Farming Systems, and Experimental Design

The study included thirty male Croatian Posavac horses, bred according to the traditional breeding practices of the Croatian Posavac breed in its native area, the Sava River basin. The typical practice for native horse breeds, such as the Croatian Posavac, involves free-range grazing on common pastures in the Sava River basin for about eight months during the growing season (March to November). Afterward, the herds (stallions, mares, and foals) are collected and housed in facilities for the next four months (November to February), where they are fed hay and a modest amount of concentrated feed (a simple mixture of corn, barley, and oats). Foals not kept for breeding or intended for meat production are fed 5 kg of concentrated feed per day (corn, barley, and oats, with about 60% corn) during the last three months before sale. This is done to ensure their good condition and favorable carcass characteristics. This research was approved by the Bioethical Committee for the Protection and Welfare of Animals in the Faculty of Agriculture, University of Zagreb (No: 251-71-29-02/19-22-2) and was conducted in agreement with the Croatian Posavina Breeders’ Association. The foals were born in March and April 2022. After birth, the foals were raised on pastures in a free-range system for approximately eight months, together with mares (mothers) so that they could suckle their mothers without interference. During the grazing period (March/April to November), no supplementation with hay or concentrates was provided. During the grazing period, foals become used to grazing, from which they cover their development and growth needs after weaning. In November, the foals were gathered and moved from the pastures to facilities where they were housed in group pens. The feeding was based primarily on hay, but during the last two months before the planned sale, a concentrated feed of 2 to 3 kg/day was introduced into the young horses’ ration. This feeding regime was maintained until the end, i.e., until the animals were loaded for transport to the slaughterhouse.

### 2.2. Carcass and Meat Characteristics

The foals were transported by truck from the farm to the slaughterhouse in under two hours. Prior to transport, the foals were fasted for 12 h, with access to drinking water. Before slaughter, the foals were weighed to determine their body weight. Slaughter was carried out in an EU-approved slaughterhouse following the specifications in the European legislation [44], following standard procedures, which included stunning with a captive bolt pistol with a penetrating rod, exsanguination by severing the large jugular veins in the hanging position, skinning, the removal of abdominal and thoracic organs, final cleaning, and chilling at 4 °C in a cold chamber for 24 h.

After chilling, the cold carcass weights of the left and right halves of each carcass were measured to estimate the dressing percentage. The carcasses were then cut on the left side at the level between the 9th and 10th thoracic vertebrae. The pH_24_ and color were measured on the cross-sectional area of the *musculus longissimus dorsi* (MLD) at the level of the 10th thoracic vertebra. The pH_24_ value was measured using Eutech CyberScan pH 310 equipment (Eutech Instruments Pte Ltd., Singapore). Color parameters (*L**, *a**, *b**, *C**, *H**) were assessed according to CIE standards [45] after a bloom time of 80 min using a Minolta Chroma Meter CR-410 (Minolta Co., Ltd., Tokyo, Japan) with a 50 mm diameter measuring area and D65 illumination. On the same cross-section, the surface area of the MLD was drawn on a transparent foil and subsequently measured using a Robotron planimeter (Reiss Precision, Bad Liebenwerda, Germany).

Subsequently, dissection of the “rib section” between the 10th and 17th thoracic vertebrae was conducted using a knife, immediately after carcass processing, on the left side of ten animals to evaluate the proportions of muscle, bone, fat, and connective tissue. The *musculus longissimus dorsi* was then portioned into eight steaks, each 2.5 cm thick, and vacuum-packaged, and six steaks were stored at −20 °C until further analysis. The first two steaks were used for sensory evaluation. The subsequent four steaks were utilized to determine the chemical composition, fatty acid and amino acid profiles, and volatile compounds of the foal meat. The remaining two steaks, which were not frozen, were used to assess shear force, drip loss, and cooking loss.

Water-holding capacity was measured through drip loss and cooking loss. To determine drip loss, a raw, non-frozen meat sample (100 to 120 g and approximately 1.5 cm thick) was placed in a DripLoss container (Christensen ApS, Hillerød, Denmark). The container was sealed and stored at 4 °C for 24 h. After this period, the meat samples were removed and weighed. Drip loss (%) was calculated as the difference in sample weight before and after exudation. Cooking loss was determined using a raw meat sample (approximately 120 g). The samples were weighed and then cooked in sealed plastic bags at 80 °C in a water bath with automatic temperature control. After cooking, the samples were cooled to room temperature (18 ± 2 °C) and weighed again. Cooking loss (%) was calculated as the difference in weight before and after cooking and cooling divided by the initial weight and multiplied by 100 [46]. Meat texture was evaluated using a TA-HDi Texture Analyser with a Warner–Bratzler shear attachment (10 N load cell; 2 mm/s), and analyzed with Texture Expert software 1.22 (Stable Micro Systems Ltd., Godalming, UK) [47].

### 2.3. Fatty Acid Analysis

The chemical analysis of the meat was performed on a 100 g sample of MLT from the 10th rib using near-infrared transmission spectrophotometry (NIRS) in the range of 850–1050 nm with a Food Scan instrument (Foss Electric A/S, Hillerød, Denmark).

The total lipids from the horse meat were extracted using a modified Folch method [48] with a chloroform/methanol solvent system extraction in volumetric ratios of 2:1, 1:1, and 1:2 (*v*/*v*). Each extraction was performed for 30 min with stirring (700 rpm), followed by centrifugation at 3000 rpm for 10 min at 20 °C. The lipid extracts were transesterified according to ISO 5509:2000 [49] to obtain fatty acid methyl esters (FAMEs), which were analyzed using gas chromatography (Agilent 8860, Agilent Technologies, Santa Clara, CA, USA) with a flame ionization detector (FID) and a DB-23 capillary column. The injector and detector temperatures were set to 200 °C and 240 °C, respectively. The oven program started at 120 °C (3 min), increased to 260 °C at 6 °C/min, and was held for 5 min. Hydrogen was used as the carrier gas (1 mL/min). FAMEs were identified by comparing retention times with standards (Sigma-Aldrich and Supelco, Merck KGaA, Darmstadt, Germany) and quantified using methyl nonadecanoate (C19:0) as the internal standard. Fatty acid composition was expressed as a percentage of the total fatty acids. Atherogenic and thrombogenic indices were calculated according to Ulbricht and Southgate [50].

### 2.4. Amino Acid Analysis

Quantitative analysis of total amino acids in the meat was performed using the high-performance liquid chromatography method (HPLC). Meat samples (2.5 g) were minced, homogenized, and then diluted with ultrapure water with the addition of an internal standard (α-aminobutyric acid) to a final concentration of 1 mM. To 100 μL of the sample, 100 μL of 12 M HCl was added, after which the samples were hydrolyzed (21 h/114 °C), lyophilized (60 °C), dissolved in 5 mL of 20 mM HCl, and filtered through a 0.45 μm pore-size filter. The derivatization of amino acids was carried out using AccQ•Fluor reagent kit according to the manufacturer’s instructions (Waters, Milford, CT, USA). The standard amino acid solution at a concentration of 2.5 mM was diluted to 100 pmol/μL. The quantitative analysis of amino acids was performed using an HPLC system with α-aminobutyric acid as the internal standard. The results were expressed as milligrams of amino acid per gram of sample (mg/g). Since Asp is converted to Asn and Glu to Gln during acid hydrolysis, they could not be quantified individually; therefore, Asp and Asn were quantified together and reported as Asx, while Glu and Gln were quantified together and reported as Glx.

### 2.5. Sensory Valorisation

Sensory evaluation was performed on five horse meat samples by a panel of eight trained assessors, following a structured sensory evaluation protocol in collaboration with the Istituto Italiano Assaggiatori Carne De Gustibus Carnis (available online: www.degustibuscarnis.it, accessed on 12 March 2025). The evaluation was conducted in two phases: the first phase focused on fresh (raw) meat, while the second involved thermally processed meat that had previously been evaluated in its fresh state. Frozen samples were thawed under refrigerated conditions (4 °C) for 24 h prior to analysis. For the evaluation of fresh meat, steaks were tempered to room temperature by holding them unpackaged for 60 min before analysis. Thermal processing involved grilling each steak on both sides until an internal temperature of 58 °C was reached. Following grilling, the meat was wrapped in aluminum foil and allowed to rest for 2 min before being portioned into pieces measuring 1.5 cm × 1.5 cm × 2.5 cm. Two randomized portions from each steak were individually wrapped in aluminum foil, labeled with corresponding three-digit codes, and presented to panelists for sensory evaluation. Sensory assessments were performed using a structured 10-point scale (1 = extremely low intensity or poor quality; 10 = extremely high intensity or excellent quality) encompassing four main sensory categories: visual perception, olfactory perception, tactile and gustatory perception, and retronasal perception.

On fresh, raw meat cuts, visual and olfactory perception were evaluated. Visual perception included the following attributes: color intensity (saturation of red), color uniformity (evenness of distribution), surface gloss (light reflectance indicating freshness), fineness of muscle fibers (thickness and visibility of individual fibers), and intramuscular fat distribution (visible marbling, ranging from slight striations to heavier deposits). Additionally, overall visual appeal was scored based on the visual attractiveness of the meat cut as they would appear at the point of purchase. Olfactory perception was assessed through direct smelling and included odor intensity (overall strength of aroma), the detection of atypical or off-odors (indicators of spoilage or processing faults, such as rancid, sulfuric, or chemical smells), and overall odor liking (general aromatic pleasantness).

On thermally processed (grilled) meat, tactile, gustatory, and retronasal perceptions were evaluated. Tactile and gustatory assessment included attributes such as tenderness, juiciness, fat perception, fibrousness, metallic taste, and bitterness. In addition, flavor liking and texture liking were scored, culminating in an overall evaluation of tactile and gustatory pleasantness. Retronasal perception, referring to aroma and taste sensations perceived after swallowing, included the perceived intensity of fresh meat, cooked meat, and roasted meat notes, as well as additional descriptors of vegetal, liver-like, or biochemical origin. Aroma persistence, overall flavor pleasantness, and total retronasal liking were also evaluated. Finally, an overall hedonic impression of the meat was assessed, reflecting the integrated satisfaction from all sensory dimensions.

### 2.6. Volatile Aromatic Compound GC-MS Analysis

For the analysis of aromatic compounds in horse meat, the gas chromatography–mass spectrometry (GC-MS) method was applied in combination with headspace solid-phase microextraction (HS-SPME). Before the analysis, the meat sample was thawed and tempered to room temperature, after which it was grilled on an electric grill to an internal temperature of 50 °C. After cooling at room temperature for 30 min, the sample was minced, and 2 g of the minced sample was placed in a 50 mL Falcon tube with 20 mL of saturated NaCl solution and homogenized. In a headspace vial, 3 mL of the prepared sample, together with 10 μL of the internal standard 2-methyl-3-heptanone (0.05 μg/μL), was subjected to volatile compound equilibration at 40 °C for 20 min under constant mixing. The adsorption of volatile compounds was carried out using an SPME fiber for 60 min at the same temperature. The analysis was performed on a Thermo Fisher GC Trace 1300 Chromatograph (Thermo Fisher Scientific, Waltham, MA, USA) coupled with a TSQ 9000 triple quadrupole mass spectrometer, using a DB-225MS column and helium as the carrier gas. With the splitless injection mode, the inlet temperature was 250 °C, while the analysis lasted 48 min at a constant flow rate of 1.5 mL/min. Volatile components were detected in Full Scan mode (*m*/*z* range 30–350) using EI ionization. Identification was performed by comparing the spectra with the NIST MS Search database (ver. 2.3). The relative abundances of the identified compounds were calculated based on the ratio of the signal of each component to the internal standard. The concentrations of the volatile aromatic compounds were expressed in micrograms per kilogram (µg/kg).

### 2.7. Statistical Analysis

Descriptive analysis was performed using Statistical Analysis Software (SAS V.18 Inc., Cary, NC, USA). All the presented tables contain the least square mean (LS Mean) and the standard error (SE) of the means.

## 3. Results

The results of the growth dynamics, slaughter, and carcass characteristics of the Croatian Posavina horse foals are presented in Table 1. The average slaughter age was 10.5 months, at which the foals reached a live weight of 347.2 kg and a cold dressing percentage exceeding 60%. Both the gross and daily carcass gains indicate a favorable growth rate.

The measured pH_24_ value of 5.67 indicates a normal post mortem process with sufficient conversion of glycogen to lactic acid. This value lies within the expected physiological range for horse meat and confirms the absence of quality defects such as PSE (pale, soft, exudative) or DFD (dark, firm, dry) meat.

The *L** value of 39.83 reflects moderate lightness, typical for horse meat from young foals. It suggests a balanced myoglobin content and supports a natural, slightly darker red appearance, which is generally well accepted by consumers seeking red meat with visual depth and freshness. High *a** values (19.47) are associated with greater oxymyoglobin presence and adequate oxygenation, both of which contribute significantly to the perceived freshness of the meat. The *b** value of 4.51 represents a mild yellow component in the color profile, adding visual warmth without compromising the dominant red tone. This degree of yellowness is common in horse meat and remains within acceptable sensory frame. The chroma (*C**) value reflects strong color saturation. Such a high chroma enhances the visual appeal of the meat, producing a vivid and intense color impression typically preferred by consumers. The hue angle (*H**) of 13.02 indicates a color profile strongly dominated by red tones, with minimal shift toward yellow. This low hue value, in combination with high redness and chroma, confirms a stable and desirable appearance, characteristic of quality horse meat.

The tissue composition of the rib sections and selected meat quality traits of the Croatian Posavina horses are presented in Table 2. The average surface area of the *m. longissimus dorsi* was 48.17 cm^2^. Shear force, used as an indicator of meat tenderness, averaged 3.92 kg/cm^2^. The drip loss and cooking loss were 3.73% and 24.35%, respectively, indicating moderate water-holding capacity and typical thermal shrinkage during heat treatment.

The high proportion of muscle tissue (>70%) and the relatively low shares of bone and fat indicate a favorable carcass yield.

The composition and fatty acid profile of the *m. longissimus dorsi* in the Croatian Posavina horses, as presented in Table 3, provide insight into the nutritional characteristics of the meat. The average protein and fat contents indicate a typical composition for horse meat, with a relatively high protein-to-fat ratio (6.2:1), which is considered favorable from a nutritional perspective.

The fatty acid profile of the *m. longissimus dorsi* in the Croatian Posavina foals reveals a balanced composition of saturated (SFAs), monounsaturated (MUFAs), and polyunsaturated fatty acids (PUFAs), highlighting the nutritional value of the meat. Among the total saturated fatty acids (SFAs), palmitic, stearic, and myristic acids were predominant, while short- and medium-chain SFAs (C8:0–C12:0) were present in low concentrations (<1%). Within the monounsaturated fatty acid (MUFA) group, oleic acid (24.10) was the most abundant, followed by palmitoleic acid. Linoleic acid was the dominant polyunsaturated fatty acid (PUFA), followed by α-linolenic acid. Long-chain PUFAs, including arachidonic and eicosapentaenoic acids, were also detected at levels of 2.47 and 1.04, respectively. The overall unsaturated fatty acid (UFA) content was 59.01, resulting in a UFA/SFA ratio of 1.44 and a PUFA/SFA ratio of 0.70, both of which were within nutritionally favorable ranges. The ω-6/ω 3 ratio was 3.46, which, although at the upper end of recommended values (≤4), still reflects a relatively balanced proportion between pro-inflammatory and anti-inflammatory fatty acids. The AA/EPA ratio (2.90) further contributes to understanding the balance of biologically active lipid mediators. The desaturase activity indices, SCDi16 (C16:1/C16:0) at 17.75 and SCDi18 (C18:1/C18:0) at 76.03, reflect the activity of the enzyme stearoyl-CoA desaturase (SCD), which plays a central role in lipid metabolism by introducing a double bond into saturated fatty acyl-CoA substrates. Higher values of these indices suggest the increased enzymatic conversion of saturated fatty acids into their monounsaturated counterparts, which can have implications for both meat quality and nutritional properties. The elevated SCDi18 value (76.03) indicates a highly efficient desaturation of stearic to oleic acid, contributing significantly to the overall MUFA content. This enzymatic activity also affects the oxidative stability, texture, and flavor of the meat.

The amino acid composition of the *m. longissimus dorsi* in the Croatian Posavina horse is presented in Table 4. The total content of essential amino acids was 70.94 mg/g, with lysine and leucine present in the highest concentrations. The total content of non-essential amino acids amounted to 88.19 mg/g, with glutamic acid/glutamine and aspartic acid/asparagine as the predominant components.

The ratio of essential to non-essential amino acids was 0.81, indicating a balanced amino acid profile with a slightly higher share of non-essential amino acids. This composition supports the nutritional quality of Croatian Posavina horse meat, making it a valuable source of both the essential and non-essential amino acids required for human metabolism and various physiological functions.

The sensory analysis of Croatian Posavina horse meat revealed overall positive evaluations across most categories. In terms of visual perception, the meat showed high scores for color intensity (7.52), color uniformity (7.87), fineness of muscle fibers (7.15), and overall visual appeal (7.72), indicating a visually attractive product. However, marbling received a notably low score (1.87), suggesting low intramuscular fat content. Olfactory perception was also favorable, with a high overall odor appeal (7.32) and low detection of atypical odors (0.57), indicating freshness and the absence of unwanted aromas. The tactile and gustatory perception of thermally processed meat showed favorable tenderness (7.82), strong texture appeal (7.77), and high flavor appeal (7.57), while negative attributes such as metallic (1.12) and bitter tastes (0.62) remained low. Low scores for fat perception (1.32) and fibrousness (2.95) reflect the lean character of the meat. In retronasal perception, roasted meat aroma scored high (6.57), while fresh (1.70) and cooked meat notes (1.75) were less prominent. Unpleasant or unusual notes—such as liver-like (1.05) and biochemical (0.30) descriptors—were minimal. The persistence of aroma and odor was rated highly (7.40), as was overall retronasal appeal (7.57). Ultimately, the overall hedonic impression reached a high score of 7.75, reflecting a strong positive response from panelists and confirming the high sensory quality of Croatian Posavina horse meat.

The analysis of volatile aromatic compounds in Croatian Posavina horse meat revealed the presence of a diverse range of chemical constituents, with notable differences in their concentrations. Among the detected compounds, 2-methyl-4-heptanone had the highest concentration (25.90), followed by (R)-1-octen-3-ol (1.90), which is often associated with mushroom-like and meaty aromas. Other compounds with relatively high levels included benzaldehyde (0.72), known for its almond-like scent, and 1,2-benzenedicarboxylic acid (0.15). Moderate levels of 1-octanol (0.14) and butanoic acid (0.13) were also observed, the latter contributing to characteristic fatty and cheesy odors.

Most of the remaining compounds were present in trace amounts, including various aldehydes, alcohols, lactones, and aromatic hydrocarbons, each contributing subtly to the overall aroma profile. The chemical composition supports the sensory findings, suggesting that specific volatiles play a role in the meat’s flavor complexity and overall sensory appeal.

## 4. Discussion

The slaughter performance indicators of Croatian Posavina foals, as presented in the current research (Table 1), rank this local horse breed within the mid-range of meat production efficiency compared to other European cold-blood breeds and crossbreeds. The dressing percentage of the Croatian Posavina horse is similar to that reported for Croatian horses by Manfredini et al. [6] (61.6%). However, when compared to high-yielding European heavy breeds such as the Italian Heavy Draught horse [11,15] (68.9%, 72.6%) and Heavy French breeds [4] (64.9%), the Croatian Posavina horse shows a lower dressing percentage and lighter carcass, which may result from genetic predisposition as well as semi-extensive production systems. Conversely, the Croatian Posavina horse performs significantly better than smaller mountain-type breeds, such as the Galician Mountain horse [14,17] (53.3%, 47.7%) and its crossbreeds with Hispano-Bretón horses [18] (50.07%). The especially low dressing percentages in these groups (often below 55%) suggest that the Croatian Posavina breed offers relatively competitive meat yield despite its smaller frame. For instance, Frenches Montagnes horses in the study by Badiani et al. [7] showed a dressing percentage of 61.7%, while other studies [8] reported higher values (68.2%). Haflinger horses also showed lower dressing values (50.6%; 59.66%) than the Croatian Posavina horse [9,13]. Crossbreeding heavy and mountain breeds, such as the Galician Mountain × Hispano-Bretón [18,19], reached 50.07% and 54.09%, indicating that increased live mass does not necessarily correlate with a higher dressing percentage. The highest dressing percentages (72.6%) were achieved with relatively young Italian Heavy Draught horses [15], suggesting that intensive breeding can enable high meat yields even at early ages. Breeds like Burguete [12] achieved 65% at 24 months. The Croatian Posavina foals achieve satisfactory dressing percentages under extensive pasture-based farming, and together with their adaptability to local ecosystems, this supports their role as a sustainable meat production option in marginal areas, especially within agroecological and traditional livestock systems.

When the tissue composition (muscle: fat: bone) analysis results from the rib section sample of the Croatian Posavina foals (Table 2) are compared to other horse breeds and their crosses, it can be seen that Croatian Posavina foals have a balanced tissue profile, characterized by well-developed musculature and moderate fat deposition, traits that are particularly important in the context of meat production. Comparable values were reported for Heavy French breeds [4] at 12 months of age, with a similar muscle percentage (70.10%) and slightly lower fat content (10.90%), along with a reduced bone content (15.60%). On the other hand, Italian Heavy Draught Horses [11,15] have significantly higher muscle contents (72.10% and 75.03%) and lower bone contents (11.90% and 9.55%). In contrast, smaller-framed or extensively raised breeds such as the Galician Mountain horse [14,16,17] show relatively high muscle values (69.3 to 73.0%) but a considerably higher bone content (20.2% to 27.2%), which reduces overall carcass yield. The fat content (ranging from 3.4% to 6.7%) observed in these animals suggests a leaner condition, likely a result of harsher environmental conditions and restricted nutritional input. Franches-Montagnes horses have lower muscle proportions (63.60–67.70%) and greater variability in fat percentages (9.50–16.40%), which may be attributed to varied management practices [5,8]. The Croatian Posavina foals show a competitive carcass tissue composition, particularly in the context of their extensive production system.

The values of drip loss and cooking loss (Table 2) indicate a moderate water-holding capacity and satisfactory tenderness of the meat from Croatian Posavina foals. In terms of drip loss, the Croatian Posavina horses had slightly higher drip and cooking losses compared to Galician breeds and their crosses. Lorenzo et al. [17] reported a drip loss of 3.10% in the Galician Mountain horse, while Domínguez et al. [19] and Franco & Lorenzo [18] found even lower values in Galician Mountain × Hispano-Bretón crosses (2.34% and 2.14%, respectively). Cooking loss in the Croatian Posavina foals is higher than that observed in Galician crosses (15.66–21.50%) [17,18,19] but lower than that reported for Italian Heavy Draught Horses (37.92%) [15]. Similarly, it is below the values recorded in Sanfratellano horses (26.57%) [13], placing the Croatian Posavina horse in the mid-range for cooking losses, which is acceptable from culinary and technological perspectives. The observed shear force value indicates moderate tenderness, a key quality trait influencing consumer acceptance. Compared to other breeds, this value is lower than that reported for the Italian Heavy Draught horse by De Palo et al. [15] at the same slaughter age (5.31 kg/cm^2^). In contrast, lower shear force values were observed in Galician Mountain horses slaughtered at 15 months, with Lorenzo et al. [17] reporting 3.80 kg/cm^2^ and Franco & Lorenzo [18] reporting 2.91 kg/cm^2^ for a Galician Mountain and Hispano-Bretón crosses. Overall, the shear force results of Croatian Posavina foal meat indicate acceptable tenderness, positioning the breed competitively in terms of meat quality within the context of extensive or semi-extensive production systems. Combined with its balanced tissue composition and acceptable carcass yield, these findings reinforce the breed’s viability for sustainable meat production and potential for inclusion in premium or regional meat markets.

Meat color is a key indicator of freshness and quality perception among consumers, directly influencing their purchasing decisions. It is affected by the structural characteristics of the muscle and physiological post mortem processes. In the present study, Croatian Posavina foal meat, evaluated 24 h post mortem, showed acceptable pH_24_ and color (*L**, *a**, *b**) values (Table 1). These results indicate normal post mortem glycolysis and a desirable color profile for the horse meat. The pH_24_ value of the Croatian Posavina foal meat is within the physiological range (5.5 to 5.7) typically associated with favorable meat quality. It is slightly higher than the values reported for other breeds such as Italian Heavy Draft horses (5.59) [23] and Galician crossbreeds (5.59–5.65) [18,19], similar to that of Polish heavy horses (5.72) [24], and comparable to Sanfratellano (5.69) and Haflinger (5.61) horses [13]. In this study, the *L** value of Croatian Posavina foal meat (38.83) suggests favorable brightness. Higher *L** values were observed by Sarriés and Beriain [22] and Litwińczuk et al. [24] in Polish heavy horses (49.54, 44.90). The Italian Heavy Draft horse [23] and Galician Mountain × Hispano-Bretón crosses [16] have lower *L** values (36.72 and 35.93, respectively). Darker meat is often linked to higher myoglobin concentration and lower intramuscular fat content, commonly associated with pasture-based, physically active animals. The Croatian Posavina horse, which is traditionally reared in extensive systems, demonstrates a comparable meat appearance to these breeds, supporting the notion that production systems and animal activity levels significantly influence meat lightness. The redness parameter (*a**) is related to myoglobin content and oxygenation state. Meat from Croatian Posavina foals achieved a value of 19.47, which is among the highest reported across all studied breeds, second only to the Catria horse (23.20) [25]. This finding implies a very intense red color, which is typically considered desirable by consumers, especially when red meat is associated with freshness and quality. By comparison, Galician Mountain breeds and their crosses generally showed significantly lower *a** values (ranging from 9.89 to 17.60) [14,17,18,19], as did Italian Heavy Draft horses (11.71) [23]. Sanfratellano and Haflinger horses also have intermediate *a** values (15.46 and 17.02, respectively) [13], confirming that Croatian Posavina meat is characterized by favorable redness, possibly due to enhanced oxidative muscle profiles and lower-stress handling pre-slaughter. Yellowness (*b**) reflects the contribution of yellow hues, which can be influenced by fat content, carotenoid intake, and muscle oxidation levels. The *b** value of 4.51 for Croatian Posavina foals suggests a mild yellow component, which is lower than in most heavy breeds such as Polish horses (19.60, 14.32) [22,24]. Compared to other extensively reared breeds like Galician Mountain crosses [14,17,18,19] (4.32 to 11.64), Croatian Posavina foal meat has a relatively lower *b** value, consistent with leaner muscle profiles and moderate fat deposition. Some studies reported negative *b** values in Italian Heavy Draft horses (−1.67) [23]. The Croatian Posavina horse meat is characterized by a favorable and balanced color profile. The intense redness, moderate brightness, and yellowness create a quality visual appearance likely to appeal to consumers. These parameters, especially when compared to other European heavy and local horse breeds, support the marketability and visual attractiveness of Croatian Posavina horse meat, particularly in niche markets or meat markets focused on traditional, pasture-raised products.

The moisture content of Croatian Posavina foal meat (Table 3) is comparable to values reported for several other horse breeds. For instance, Burguete horse meat has a similar moisture content of 72.32% [12], while the Sanfratellano breed shows slightly higher values at 73.23% [13]. In comparison, Galician Mountain horses have higher moisture levels, around 75% [14]. Although slightly lower than in Galician Mountain horses, the moisture content in Croatian Posavina foals aligns with that reported for Italian Heavy Draft horses (69.51%) [23] and Polish heavy horses (69.78%) [24]. The protein content in Croatian Posavina foal meat (22.37%) is comparable to that of Galician Mountain horses, whose protein content ranges from 20.44% to 22.30% [14,17]. Higher protein contents have been reported in Sanfratellano and Haflinger horses, at 23.09% and 23.19% [13], respectively. Compared to data from Makray et al. [9], who reported a protein content of 21.20% in heavy horses, Croatian Posavina meat shows slightly higher values. Its protein content is also noticeably higher than that of Polish heavy horses, which contain 19.67% protein [24]. The fat content in Croatian Posavina foal meat indicates a moderate level of intramuscular fat (IMF) compared to other breeds. Polish heavy horses have a higher fat content (6.59%) [24], suggesting increased marbling, while Italian Heavy Draft horses also show higher fat levels, averaging 4.31% [23]. Conversely, Galician Mountain and Catria horses show significantly lower IMF values, ranging from 0.16% to 2.83% [14,25]. Makray et al. [26] reported fat contents in heavy horses ranging from 1.9% to 3%, suggesting leaner meat profiles. The ash content in Croatian Posavina foal meat is 1.09%, consistent with other draft breeds. For example, Polish heavy horses and Burguete horses present similar ash contents at 1.10% and 1.13% [12,24], respectively. Slightly higher ash levels have been reported in the Sanfratellano (1.39%) and Italian Heavy Draft horse (1.38%) [13,15]. In contrast, Galician Mountain horses have notably lower ash contents, ranging from 0.16% to 0.36% [14]. Overall, Croatian Posavina foals have a balanced chemical meat composition in terms of protein, fat, and ash, indicating the breed’s potential to provide a consistent and nutritionally valuable meat product.

The intramuscular fatty acid (FA) profile of Croatian Posavina foals largely corresponds with previous studies. Among the saturated fatty acids (SFAs), palmitic acid (C16:0) was predominant at 27.27 g/100 g of total lipids, which is comparable to the level in Galician Mountain foals (28.65 g/100 g) [29] and Haflinger foals (26.16 g/100 g) [13]. The stearic acid (C18:0) level in Croatian Posavina foals was 7.6 g/100 g, higher than in Galician foals (4.81 g/100 g) [29] and similar to Sanfratellano (6.43 g/100 g) [13]. Regarding monounsaturated fatty acids (MUFAs), oleic acid (C18:1 c9) in Croatian Posavina foals reached 24.1 g/100 g, which is slightly lower than in Galician Mountain horses (30.35 g/100 g) [29], Catria foals (31.14 g/100 g) [25], and Galician × Hispano-Bretón crosses (30.18 g/100 g) [19]. Palmitoleic acid (C16:1 9c) was 5.93 g/100 g, comparable to Sanfratellano (5.45 g/100 g) and Haflinger (5.13 g/100 g) [13], but lower than in Italian Heavy Draught Horses (9.75 g/100 g) [30]. In terms of polyunsaturated fatty acids (PUFAs), linoleic acid (C18:2 n−6) was the dominant PUFA in Croatian Posavina foals (18.93 g/100 g, present study), higher than in Galician × Hispano-Bretón (13.19 g/100 g) [19] and Catria foals (7.52 g/100 g) [25], and comparable to Haflinger (20.99 g/100 g) [13]. Of particular nutritional interest is the ω-3 PUFA fraction. The level of α-Linolenic acid (C18:3 ω-3) in Croatian Posavina foals was 6.06 g/100 g, a value that exceeds that in Sanfratellano and Haflinger (1.88 g/100 g, 3.90 g/100 g) [13] and is close to that of Galician Mountain foals (7.46 g/100 g) [29]. Even higher concentrations were found in Galician × Hispano-Bretón crosses (7.66 g/100 g) [19] and Galician Mountain and Hispano-Bretón foals (11.64 g/100 g) [18]. Additionally, eicosapentaenoic acid (EPA, C20:5 ω-3) was found at 1.04 g/100 g in Croatian Posavina foals, surpassing reported values in Italian Heavy Draft breeds (0.03 g/100 g) [30] and similar to Catria foals (0.19 g/100 g) [25]. The level of arachidonic acid (C20:4 n−6) in Croatian Posavina foals was 2.47 g/100 g, notably higher than in Galician foals (0.63 g/100 g) [29] and Galician Mountain and Hispano-Bretón crosses (0.21 g/100 g) [18]. Other long-chain PUFAs such as docosahexaenoic acid (DHA, C22:6 ω-3) were found at 0.05 g/100 g in Galician Mountain foals [29], and at slightly higher levels in Sanfratellano and Haflinger breeds (0.37 g/100 g, 0.40 g/100 g) [13].

Among the various lipid fractions, saturated fatty acids (SFAs), polyunsaturated fatty acids (PUFAs), and unsaturated/saturated ratios (UFA/SFA and PUFA/SFA) are the most relevant indicators for assessing the atherogenic risk of meat. In the present study on Croatian Posavina foals, the SFA content was 40.99%, which is moderately high compared to other breeds (e.g., 36.86% in Sanfratellano and 35.88% in Galician Mountain × Hispano-Bretón at 15 months [13,18]). However, what distinguishes Croatian Posavina foals is the exceptionally high PUFA content (28.5%), which ranks among the highest reported across all examined groups, surpassed only by Galician Mountain and Hispano-Bretón foals at 15 months (32.61%) [18]. High PUFA levels contribute to a favorable PUFA/SFA ratio, which in Croatian Posavina foals is 0.70, placing them ahead of many traditional heavy breeds such as the Italian Heavy Draught horse (0.57) [15], the Catria breed (0.23) [25], and Italian Heavy Draft horses (0.44) [30]. Only a few crossbred or mountain-adapted lines, such as Galician Mountain and Hispano-Bretón (0.91) [18], exceeded this value. The UFA/SFA ratio is another critical nutritional index, reflecting the balance between health-promoting unsaturated fats and atherogenic saturated fats. Croatian Posavina foals demonstrate a UFA/SFA ratio of 1.44, which is comparable to values reported in other breeds such as Italian Heavy Draft horses (1.47) [30], though slightly below high-performing types such as the Haflinger (1.75) [13] and Galician crosses (1.79) [18]. These findings suggest that the meat of Croatian Posavina foals has a well-balanced lipid profile, with a particularly advantageous PUFA content, distinguishing it from heavier, more intensively reared breeds such as the Catria or Italian Heavy Draught horse [25,30], which show higher SFA and lower PUFA levels. This favorable composition may be attributed to extensive rearing practices, forage-based diets, and the slower growth typical of Croatian Posavina foals, aligning with similar nutritional advantages observed in other horse breeds [19,29]. This positions Croatian Posavina foal meat as a valuable alternative for health-conscious consumers seeking red meat with enhanced functional properties.

The fatty acid (FA) profile of foal meat plays a significant role in its nutritional value, particularly concerning human health, due to its influence on cardiovascular disease risk. Special attention is also given to the ω-6/ω-3 ratio, which serves as a biomarker for the balance of pro-inflammatory and anti-inflammatory compounds in human health. According to current research, the meat of Croatian Posavina foals contains 21.4 ω-6 PUFAs and 7.1 ω-3 PUFAs, resulting in an ω-6/ω-3 ratio of 3.46. These results highlight a relatively balanced PUFA composition, with a favorable presence of ω-3 fatty acids compared to most other breeds. For example, Sanfratellano foals at 18 months show a higher total ω-6 content (24.82) but a significantly lower ω-3 content (4.18), yielding a less favorable ω-6/ω-3 ratio of 5.93 [13]. Similarly, Haflinger foals present an ω-6/ω-3 ratio of 3.77 [13], indicating a less optimal balance compared to Croatian Posavina foals. Galician Mountain × Hispano-Bretón foals and Galician Mountain and Hispano-Bretón foals have more favorable ω-6/ω-3 ratios, at 1.78 and 1.33 [18,19], respectively, likely reflecting both genetic and environmental influences. Croatian Posavina foals position themselves in the upper-middle range in terms of PUFA quality, and their relatively high ω-3 PUFA content (7.1) provides a nutritional advantage, especially when compared to the Catria breed, which shows a ω-3 level of only 2.84 [25], and Italian Heavy Draught horses (4.5) [15]. The optimal human dietary ratio of ω-6 to ω-3 PUFAs is generally recommended to be below 4:1, with lower values considered more beneficial in preventing chronic inflammation and cardiovascular diseases. In this context, the ω-6/ω-3 ratio of 3.46 in Croatian Posavina foals places their meat within the recommended nutritional guidelines. Croatian Posavina foals thus provide a meat profile that aligns closely with health-oriented dietary recommendations, supporting their potential for premium niche markets emphasizing functional and nutritionally superior red meat.

The Atherogenic Index (AI) value for Croatian Posavina foal meat (Table 3) indicates a moderate cardiovascular health profile that is comparable to or more favorable than several traditional horse breeds. This value is slightly higher than that of Galician Mountain foals (0.71) [29] and lower than Sanfratellano (0.60) and Haflinger foals (0.59) at 18 months of age [13], both of which show more favorable AI profiles. Compared to Italian heavy breeds, Croatian Posavina foals have a more favorable lipid profile: the AI is lower than in Italian Heavy Draught horses (1.00) [15], Italian Heavy Draft horses (0.85) [30], and Catria foals (0.95) [25]. The Thrombogenic Index (TI) value for Croatian Posavina horse meat is 0.82, which remains below the values reported for Italian heavy breeds such as the Italian Heavy Draft horse (1.37), the Italian Heavy Draught horse (1.17), and Catria foals (1.32) [15,25,30], indicating a lower thrombogenic potential. When compared to other breeds, the TI of Croatian Posavina foals is slightly higher than that of Galician Mountain foals (0.74), Sanfratellano (0.86), and Haflinger foals (0.74) [13,29]. Particularly favorable TI values have been observed in Galician Mountain × Hispano-Bretón (0.67) and Galician Mountain and Hispano-Bretón foals (0.49) [18,19], which are known for their high PUFA content and extensive grazing systems. These findings reinforce the nutritional value of Croatian Posavina foal meat. Their moderate saturated fatty acid levels, high PUFA content, and balanced ω-6/ω-3 ratio (3.46), combined with favorable AI (0.74) and TI (0.82) values, suggest that this meat could be promoted as a healthier red meat alternative. This is especially relevant in systems aimed at combining animal biodiversity conservation with sustainable and functional food production strategies. Croatian Posavina foals show an SCDi16 value of 17.75 and an SCDi18 value of 76.03, suggesting moderately active desaturation. Compared to other breeds, the SCDi16 value of Croatian Posavina foals is similar to those observed in Galician Mountain foals (17.86) and Haflinger foals (16.91), and higher than in Sanfratellano (15.11) and Galician Mountain × Hispano-Bretón foals (16.15) [13,19,29]. However, it is notably lower than that of the Italian Heavy Draught horse, which reaches 24.48 [15]. In terms of SCDi18, which reflects the conversion of stearic acid to oleic acid, Croatian Posavina foals show moderately high activity. This value is slightly below that of Haflinger (80.04) and Sanfratellano foals (79.39) [13], and lower than in Galician Mountain foals (86.32) and Galician Mountain × Hispano-Bretón foals (86.20) [19,29]. The Italian Heavy Draught horse displays the highest SCDi18 value (88.78) [30]. The observed moderate desaturase activity, combined with a favorable PUFA profile and lipid indices, contributes positively to the overall nutritional quality of Croatian Posavina foal meat.

The amino acid profile of Croatian Posavina horse meat (Table 4) shows a nutritionally valuable composition, particularly in terms of essential amino acids (EAAs). The total EAA content in Croatian Posavina horse meat reached 70.94 mg/g of meat, which is somewhat lower than the values reported by Marino et al. [30] for Italian Heavy Draft foals (84.84 mg/g) and Domínguez et al. [19] for Galician Mountain × Hispano-Bretón foals of the same age (79.13 mg/g). Among the EAAs, lysine was the most abundant in Croatian Posavina meat (14.93 mg/g), followed by leucine (13.67 mg/g) and valine (9.16 mg/g). While the lysine concentration was considerably lower than in Italian Heavy Draft foals (30.55 mg/g) [30] and Galician Mountain × Hispano-Bretón foals (17.36 mg/g) [19], the valine and leucine values were relatively comparable to those reported by Domínguez et al. [19] (9.02 mg/g and 15.95 mg/g, respectively). Interestingly, methionine content in Croatian Posavina meat exceeded that in the Galician crossbreed studied by Domínguez et al. [19], who reported only 1.31 mg/g. For non-essential amino acids (NEAAs), the total in Croatian Posavina meat reached 88.19 mg/g, which was slightly lower than the values reported by Marino et al. [30] for Italian Heavy Draft horses (90.26 mg/g) and more notably lower than that values reported in Domínguez et al. [19] (103.28 mg/g). The most abundant NEAAs in Croatian Posavina meat were glutamic acid/glutamine (25.13 mg/g), aspartic acid/asparagine (15.51 mg/g), and arginine (11.89 mg/g), aligning with general trends across all studies. The essential to non-essential amino acid ratio in Croatian Posavina horse meat was 0.81, which is close to the ratio reported by Domínguez et al. [19] (0.77) and lower than that reported by Marino et al. [30] (0.94). Interestingly, this ratio was higher than that reported for Galician Mountain horses (0.85) and Galician Mountain and Hispano-Bretón crosses (1.07) [18,33]. Croatian Posavina horse meat has a well-balanced amino acid profile and essential and nonessential fractions. These findings support the viability of Croatian Posavina horse meat as a quality protein source within alternative and sustainable meat production systems.

The sensory evaluation of Croatian Posavina horse meat revealed a generally high-quality profile. The meat displayed favorable visual appeal, with high scores for color intensity, uniformity, and muscle fiber fineness, while the low marbling score reflected its characteristically lean composition (Table 5). Olfactory and gustatory attributes were also favorable, with high ratings for overall odor, tenderness, texture, and flavor, and minimal detection of undesirable notes such as bitterness or a metallic taste. These sensory characteristics are closely linked to the meat’s chemical composition. The low intramuscular fat content contributes to its tactile properties and lean flavor profile, while the composition and concentration of volatile aromatic compounds significantly influence aroma and retronasal perception. While taste refers to the five basic modalities—sweet, sour, salty, bitter, and umami—odor arises from the aroma elicited by specific volatile compounds [51]. Raw meat is weakly flavored, typically described as salty, metallic, and rare (bloody), with a slightly sweet aroma. However, it serves as a rich source of precursors for volatile compounds [52,53]. Heat treatment initiates a cascade of reactions that lead to the development of the characteristic flavor of cooked meat. In Croatian Posavina horse meat, volatile aromatic compound analysis revealed the presence of numerous aroma-active substances (Table 6). A higher concentration of 2-Methyl-4-heptanone, a ketone compound, was observed. It imparts a mild nutty, fruity, and slightly fatty flavor [54,55], contributing to the overall flavor complexity with subtle nutty or fruity undertones. The alcohol group includes 1-Hexanol, which imparts aromas reminiscent of cut grass, chemical–winey notes, fatty and fruity tones, and a mild metallic scent [56], contributing a sense of freshness to raw meat. Among esters, γ-Butyrolactone provides a creamy, pleasant, and sweet note, and butanoic acid, a product of lipolysis, introduces a mildly cheesy nuance that enhances flavor complexity [57]. Benzaldehyde enriches the olfactory profile with almond, bitter almond, and burnt sugar notes, and lipid oxidation products such as 1-octen-3-ol and 1-octen-3-one are responsible for mushroom-like aromas [58]. Hexanal, octanal, nonanal, 2-nonenal, and 2-decenal contribute fruity–floral, vegetable, herbaceous, and/or chemical notes [59]. The aldehydes (2E,4E)-Nonadienal and (2E,4E)-Decadienal are key contributors to the roasted meat aroma, while decanal imparts a pleasant, fresh, and fruity–fatty note to cooked meat.

## 5. Conclusions

The Croatian Posavina horse has notable potential as a sustainable source of high-quality red meat within pasture-based production systems. Despite being a traditionally marginalized local breed, it achieves competitive carcass yields and favorable meat quality indicators, including high protein content, low intramuscular fat, and a desirable fatty acid profile. The meat is rich in polyunsaturated fatty acids, with a balanced ω-6/ω-3 ratio and essential-to-non-essential amino acid ratio. Sensory evaluation confirmed positive consumer-relevant attributes such as tenderness, flavor, and visual appeal. When compared to other European horse breeds, the Croatian Posavina foal has a balanced combination of technological, nutritional, and sensory meat properties. These findings support repositioning the breed as a viable alternative in local and health-oriented meat markets. Furthermore, promoting its economic valorization through sustainable meat production may contribute to the long-term conservation of this economically, culturally, and ecologically important breed. Future research should consider an increased number of analytical samples and focus on consumer perception and branding strategies to further strengthen the commercial potential and conservation impact of the Croatian Posavina horse.

## Figures and Tables

**Table 1 animals-15-01911-t001:** Slaughter and carcass characteristics of Croatian Posavina foals (n = 30).

Parameters		LS Mean ± SE
Age, days		321.1 ± 6.15
Live weight, kg		347.2 ± 6.33
Gross daily gain, g/day		970.1 ± 14.97
Cold carcass weight, kg		215.2 ± 3.16
Carcass gain, g/day		603.8 ± 8.51
Cold dressing carcass, %		60.62 ± 0.24
pH_24_		5.67 ± 0.03
Color	*L**	39.83 ± 1.02
	*a**	19.47 ± 0.35
	*b**	4.51 ± 0.33
	*C**	20.00 ± 0.37
	*H**	13.02 ± 0.88

**Table 2 animals-15-01911-t002:** Carcass and meat quality characteristics of Croatian Posavina foals (n = 10).

Parameters		LS Mean ± SE
Surface area of *m. longissimus dorsi*, cm^2^		48.17 ± 1.920
Shear force, kg/cm^2^		3.92 ± 0.291
Drip loss, %		3.73 ± 0.103
Cooking loss, %		24.35 ± 0.316
Tissue composition of rib sections, %	Muscle	70.12 ± 0.186
	Fat	12.02 ± 0.344
	Bone	17.86 ± 0.218

**Table 3 animals-15-01911-t003:** Chemical composition and fatty acid profile of *m. longissimus dorsi* in Croatian Posavina foals (n = 10).

Parameters	LS Mean ± SE
Moisture, %	72.08 ± 0.389
Protein, %	22.37 ± 0.169
Fat, %	3.61 ± 0.483
Ash, %	1.09 ± 0.029
Fatty acids (g/100 g of total lipids)	
C8:0	0.19 ± 0.044
C9:0	0.51 ± 0.072
C10:0	0.25 ± 0.034
C12:0	0.87 ± 0.135
C14:0	3.83 ± 0.298
C15:0	0.47 ± 0.054
C 16:0	27.27 ± 0.330
C 16:1c	5.93 ± 0.447
C 18:0	7.60 ± 0.521
C 18:1 c9	24.10 ± 0.547
C 18:2 ω-6	18.93 ± 1.286
C 18:3 ω-3	6.06 ± 1.051
C20:1	0.48 ± 0.067
C20:4 ω-6	2.47 ± 0.277
C20:5 ω-3	1.04 ± 0.139
SFA	40.99 ± 0.292
MUFA	30.50 ± 0.842
PUFA	28.50 ± 0.883
UFA	59.01 ± 0.292
UFA/SFA	1.44 ± 0.017
PUFA/SFA	0.70 ± 0.023
AA/EPA	2.90 ± 0.813
ω-6	21.40 ± 1.460
ω-3	7.10 ± 0.973
ω-6/ω-3	3.46 ± 0.572
AI	0.74 ± 0.027
TI	0.82 ± 0.039
SCDi16	17.75 ± 1.094
SCDi18	76.03 ± 1.592

SFA, Sum of Saturated Fatty Acids; MUFA, Sum of Monounsaturated Fatty Acids; PUFA, Sum of Polyunsaturated Fatty Acids; UFA, Sum of MUFA and PUFA; UFA/SFA, Ratio of UFA to SFA; PUFA/SFA, Ratio of PUFA to SFA; AA/EPA, Ratio of Arachidonic Fatty Acid to Eicosapentaenoic Fatty Acid; ω-6, Sum of Omega-6 Fatty Acids; ω-3, Sum of Omega-3 Fatty Acids; ω-6/ω-3, Ratio of ω-6 to ω-3; AI, Atherogenicity Index, AI = (C12:0 + 4 × C14:0 + C16:0)/(MUFA + PUFA); TI, Thrombogenicity Index, TI = (C14:0 + C16:0 + C18:0)/[(0.5 × MUFA) + (0.5 × PUFA ω-6) + (3 × PUFA ω-3) + (PUFA ω-3/PUFA ω-6)]; SCDi16, Stearoyl-CoA Desaturase Index for C16 fatty acids, SCDi-16 = [C16:1/(C16:1 + C16:0)] × 100; SCDi18, Stearoyl-CoA Desaturase Index for C18 fatty acids, SCDi-18 = [C18:1/(C18:1 + C18:0)] × 100.

**Table 4 animals-15-01911-t004:** Amino acid profiles of *m. longissimus dorsi* in Croatian Posavina foals (mg/g; n = 10).

Essential Amino Acids	LS Mean ± SE	Non-Essential Amino Acids	LS Mean ± SE
Lysine (Lys)	14.93 ± 0.768	Glutamic acid/glutamine (Glx)	25.13 ± 1.272
Leucin (Leu)	13.67 ± 0.795	Aspartic acid/asparagine (Asx)	15.51 ± 0.884
Valin (Val)	9.16 ± 0.481	Arginine (Arg)	11.89 ± 1.048
Isoleucine (Ile)	8.60 ± 0.484	Alanine (Ala)	9.77 ± 0.647
Threonine (Thr)	7.21 ± 0.532	Glycine (Gly)	7.96 ± 0.369
Histidine (His)	7.06 ± 0.363	Proline (Pro)	7.21 ± 0.502
Phenylalanine (Phe)	6.79 ± 0.398	Serine (Ser)	5.48 ± 0.541
Methionine (Met)	3.52 ± 0.624	Tyrosine (Tyr)	5.24 ± 0.768
∑ Essential amino acids	70.94 ± 3.636	∑ Non-essential amino acids	88.19 ± 5.153

Asx = aspartic acid (Asp) and asparagine (Asn); Glx = glutamic acid (Glu) and glutamine (Gln).

**Table 5 animals-15-01911-t005:** Sensory characteristics of Croatian Posavina foal meat (*m. longissimus dorsi*; n = 5).

	Parameters	LS Mean ± SE	95% Confidence Interval
Visual Perception (*Fresh Meat*)	Color Intensity (Saturation)	7.52 ± 0.129	7.263–7.787
	Color Uniformity	7.87 ± 0.121	7.630–8.120
	Surface Gloss	6.45 ± 0.133	6.181–6.719
	Fineness of Muscle Fibers	7.15 ± 0.123	6.901–7.399
	Marbling	1.87 ± 0.145	1.581–2.169
	*Overall Visual Appeal*	*7.72 ± 0.114*	*7.494–7.956*
Olfactory Perception (*Fresh Meat*)	Odor Intensity	6.60 ± 0.149	6.299–6.901
	Atypical (Unexpected) Odor	0.57 ± 0.108	0.356–0.794
	*Overall Odor Appeal*	*7.32 ± 0.147*	*7.028–7.622*
Tactile and Gustatory Perception (*Grilled Meat*)	Tenderness	7.82 ± 0.146	7.530–8.120
	Juiciness	7.10 ± 0.183	6.729–7.471
	Fat Perception	1.32 ± 0.147	1.028–1.622
	Fibrousness	2.95 ± 0.120	2.708–3.192
	Metallic Taste	1.12 ± 0.121	0.880–1.370
	Bitter Taste	0.62 ± 0.106	0.409–0.841
	Texture Appeal	7.77 ± 0.118	7.536–8.014
	Flavor Appeal	7.57 ±0.143	7.285–7.865
	*Overall Tactile and Flavor Appeal*	*7.67 ± 0.125*	*7.422–7.928*
Retronasal Perception (*Grilled Meat*)	Fresh Meat Appeal	1.70 ± 0.109	1.479–1.921
	Cooked Meat	1.75 ± 0.087	1.575–1.925
	Roasted Meat	6.57 ± 0.177	6.216–6.934
	Vegetal Note	2.00 ± 0.140	1.716–2.284
	Liver Note	1.05 ± 0.114	0.818–1.282
	Biochemical Note	0.30 ± 0.074	0.150–0.450
	Aroma and Odor Persistence	7.40 ± 0.117	7.163–7.637
	*Overall Aroma and Odor Appeal*	*7.57 ± 0.111*	*7.351–7.799*
Overall Hedonic Impression		7.75 ± 0.105	7.537–7.963

**Table 6 animals-15-01911-t006:** Volatile aromatic compounds in Croatian Posavina foal meat (µg/kg; n = 5).

Chemical Compound	Formula	LS Mean ± SE
γ-Butyrolactone	C_4_H_6_O_2_	0.047 ± 0.008
Butanoic acid	C_4_H_8_O_2_	0.129 ± 0.005
Benzaldehyde	C_7_H_6_O	0.718 ± 0.033
2-Cyclopropylbutane	C_7_H_14_	0.019 ± 0.004
3-Methylenecyclopentanecarbonitrile	C_7_H_9_N	0.015 ± 0.001
1,3-Diacetoxypropan-2-ol	C_7_H_12_O_5_	0.013 ± 0.001
Cyclohexene	C_8_H_12_	0.017 ± 0.005
Benzeneacetaldehyde	C_8_H_8_O	0.018 ± 0.004
3,5-Octadien	C_8_H_12_O	0.037 ± 0.006
2-Methyl-4-heptanone	C_8_H_16_O	25.905 ± 0.067
(R)-1-Octen-3-ol	C_8_H_16_O	1.904 ± 0.074
2-Octen-1-ol	C_8_H_16_O	0.023 ± 0.005
1-Hexanol	C_8_H_18_O	0.035 ± 0.006
1-Octanol	C_8_H_18_O	0.141 ± 0.014
3-Ethylbenzaldehyde	C_9_H_10_O	0.021 ± 0.001
4-Ethylbenzaldehyde	C_9_H_10_O	0.017 ± 0.004
(2E,4E)-Nonadienal	C_9_H_14_O	0.016 ± 0.002
(2E,4E)-Decadienal	C_10_H_16_O	0.022 ± 0.003
Decanal	C_10_H_20_O	0.024 ± 0.002
3,4-(Methylenedioxy) Mandelic Acid	C_9_H_8_O_6_	0.045 ± 0.004
1,2-Benzenedicarboxylic Acid	C_16_H_22_O_4_	0.151 ± 0.011
2,2,4-Trimethyl-1,3-pentanediol Diisobutyrate	C_16_H_30_O_4_	0.239 ± 0.016

## Data Availability

The data presented in this study are available on request from the corresponding author. The data are not publicly available to preserve privacy of the data.

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
