# Peer review of "Meat Production Potential of Local Horse Breeds: Sustainable Conservation Through Valorization"

_animals, 2025, doi:10.3390/ani15131911_

Round 1

Reviewer 1 Report

Comments and Suggestions for Authors

General comments

The article examines the potential for meat production of the Croatian Posavina horse breed, highlighting its nutritional quality as a way to ensure long-term conservation. The study integrates multidisciplinary analysis of zootechnical, nutritional, biochemical, and sensory evaluations. The main limitation is the sample size (10 animals) used for carcass composition and biochemical and sensory analyses (5 animals), which might limit statistical generalizability. The authors should improve the results and discussion sections, without repeating all the table results.

Specific comments:

Line 60-62 and lines 66-67 - repeated ideas, please rephrase.

Line 74-76: substantiate the statement.

The Introduction section should include the horse meat consumption habits in Croatia.

Line 108: About animals: Authors should characterize this horse breed for adult weight, adult size, etc.

Line 143: Please specify the precise site where pH was measured.

Line 151-152: Please describe the dissection procedures. It was done with a scalpel or knife? The dissection was done immediately after carcass dressing? Why were subcutaneous and intermuscular fats not trimmed separately?

Line 207: It would be interesting to have a consumer panel evaluating horse meat samples.

Line 268-273: The authors should present a summary of the results and not a repetition of Table 1.

Line 303-304: Do not repeat Table 2 values.

Line 307-311: Do not repeat Table 3 values

Line 323-336: The authors should present a summary of the main values.

Line 350-357: Do not repeat the values presented in Table 4.

Line 444: Do not repeat values; indicate the Table number where the results are presented.

Line 472: Do not repeat values; indicate the Table number where the results are presented.

The entire results section lacks a reference to the table number in which the results are presented.

Line 702. Conclusions Section: The authors should consider future research with a larger sample of horses.

Author Response

Dear Reviewer,

Thank you for your valuable suggestions. We have accepted your proposals. Below, we present your comments and our responses in the order in which they were given.

Comment, suggestion of the Reviewer 1. The article examines the potential for meat production of the Croatian Posavina horse breed, highlighting its nutritional quality as a way to ensure long-term conservation. The study integrates multidisciplinary analysis of zootechnical, nutritional, biochemical, and sensory evaluations. The main limitation is the sample size (10 animals) used for carcass composition and biochemical and sensory analyses (5 animals), which might limit statistical generalizability. The authors should improve the results and discussion sections, without repeating all the table results.

Author's response to the reviewer:

Dear Reviewer,

Thank you for your comment. We understand the weakness of the proposed manuscript related to the smaller number of samples included in the chemical analysis of meat, particularly in the sensory analysis and the analysis of aromatic compounds. During the organization of the research, we considered that, due to the limited available resources (which inevitably shape the framework of the research to some extent), it would be preferable to perform a comprehensive analysis of all chemical components, including aromas in meat and sensory evaluation, so that the observations could contribute to a better understanding of taste during consumption. Of course, we agree that future research should be perform on a larger number of samples, with a broader sensory evaluation panel, including consumers. We have added this statement at the end of the Conclusion chapter.

Specific comments:

Suggestion of the Reviewer 1: Line 60-62 and lines 66-67 - repeated ideas, please rephrase.

Author's response to the reviewer: Thank you for your comment. The sentence that repeats the same idea (Lines 66 – 67) has been deleted.

Suggestion of the Reviewer 1: Line 74-76: substantiate the statement.

Author's response to the reviewer: Thank you for your comment. We accepted your suggestion. Substantiate the statement has been added (Line 80-83) A new reference has also been added [3]: Agnoli, L.; Capitello, R.; De Salvo, M.; Longo, A.; Boeri, M. Food Fraud and Consumers’ Choices in the Wake of the Horsemeat Scandal. Br. Food J. 2016, 118, 1898–1913. doi:10.1108/bfj-04-2016-0176.

Suggestion of the Reviewer 1: The Introduction section should include the horse meat consumption habits in Croatia.

Author's response to the reviewer: Thank you for your comment. We added a specific statement and reference (Line 74-76; Dobranić et al., 2009, [2])

Suggestion of the Reviewer 1: Line 108: About animals: Authors should characterize this horse breed for adult weight, adult size, etc.

Author's response to the reviewer: Thank you for your comment.  In order to avoid overburdening the main text of the manuscript, we have added a Supplement containing a description of the breed and accompanying photographs.

Suggestion of the Reviewer 1: Line 143: Please specify the precise site where pH was measured.

Author's response to the reviewer: Thank you for your comment.  The position where pH24 was measured was added (Line 152).

Suggestion of the Reviewer 1: Line 151-152: Please describe the dissection procedures. It was done with a scalpel or knife? The dissection was done immediately after carcass dressing? Why were subcutaneous and intermuscular fats not trimmed separately?

Author's response to the reviewer: Thank you for your comment. We have added a sentence explaining the dissection procedure (Line 160). In our previous research on the assessment of muscle, bone, and fat tissue proportions - primarily in beef cattle and donkeys - we did not separately trim subcutaneous and intermuscular fat. We consider this recommendation well-founded and will apply it in future studies. However, we cannot correct this in the proposed manuscript.

Suggestion of the Reviewer 1: Line 207: It would be interesting to have a consumer panel evaluating horse meat samples.

Author's response to the reviewer: Thank you for your comment. We definitely plan to continue this line of research on horse meat evaluation, and future studies will include a consumer panel for the assessment of horse meat samples.

Suggestion of the Reviewer 1: Line 268-273: The authors should present a summary of the results and not a repetition of Table 1.

Author's response to the reviewer: Thank you for comment. We accepted your suggestion (Lines 278-280).

Suggestion of the Reviewer 1: Line 303-304: Do not repeat Table 2 values.

Author's response to the reviewer: Thank you for comment. We have accepted your suggestion (Lines 314–315), and the earlier repetition of the sentence (Lines 310–312) has been deleted.

Suggestion of the Reviewer 1: Line 307-311: Do not repeat Table 3 values

Author's response to the reviewer: Thank you for comment. We accepted your suggestion (Lines 321-323).

Suggestion of the Reviewer 1: Line 323-336: The authors should present a summary of the main values.

Author's response to the reviewer: Thank you for comment. We accepted your suggestion (Lines 340-346).

Suggestion of the Reviewer 1: Line 350-357: Do not repeat the values presented in Table 4.

Author's response to the reviewer: Thank you for your comment. It has been accepted and implemented accordingly (Lines 372-376).

Suggestion of the Reviewer 1: Line 444: Do not repeat values; indicate the Table number where the results are presented.

Author's response to the reviewer: Thank you for your comment. It has been accepted and implemented (Line 471).

Suggestion of the Reviewer 1: Line 472: Do not repeat values; indicate the Table number where the results are presented. The entire results section lacks a reference to the table number in which the results are presented.

Author's response to the reviewer: Thank you for your comment. In response to the suggestion, we have inserted the corresponding table numbers where applicable. (Line 425-426,453, 498, 536, 549, 640, 674, 702,715)

Suggestion of the Reviewer 1: Line 702. Conclusions Section: The authors should consider future research with a larger sample of horses.

Author's response to the reviewer: Thank you for your valuable suggestion. It has been accepted and implemented (Line 744).

Reviewer 2 Report

Comments and Suggestions for Authors

I really liked the manuscript, congratulations to the authors.

The number of horses included in the studies could perhaps have been greater, and the materials and methods section could have been expanded.

The results were interesting, the conclusions modest.

I can support the manuscript even without modification.

Author Response

Dear Reviewer,

Thank you for your valuable suggestions. Below, we present your comments and our responses in the order in which they were given.

Comment, suggestion of the Reviewer 2.

I really liked the manuscript, congratulations to the authors.

The number of horses included in the studies could perhaps have been greater, and the materials and methods section could have been expanded.

The results were interesting, the conclusions modest.

I can support the manuscript even without modification.

Suggestion of the Reviewer 2: The number of horses included in the studies could perhaps have been greater, and the materials and methods section could have been expanded.

Author's response to the reviewer: Thank you for your comment. We definitely plan to continue this line of research on horse meat evaluation, and future studies will include a consumer panel for the assessment of horse meat samples.

Round 2

Reviewer 1 Report

Comments and Suggestions for Authors

Dear Authors,

The manuscript was significantly improved, and the questions have been answered.

Lines 706-708: Future research should consider an increased number of analytical samples and focus on consumer perception and branding strategies to further strengthen the commercial potential and conservation impact of the Croatian Posavina horse.

Author Response

REVIEWER 1

Comment, suggestion of the Reviewer 1.

Lines 706-708: Future research should consider an increased number of analytical samples and focus on consumer perception and branding strategies to further strengthen the commercial potential and conservation impact of the Croatian Posavina horse.

Author's response to the reviewer:

Dear Reviewer,

Thank you for your comment. As suggested, we have revised the last sentence of the Conclusion Chapter (Lines 706–708; or Lines 741–743 in the track changes version)
